# A qualitative inquiry into implementing an electronic health record system (SmartCare) for prevention of mother-to-child transmission data in Zambia: a retrospective study

Sehlulekile Gumede-Moyo,[1,2] Jim Todd,[1] Virginia Bond,[1,3] Paul Mee,[1] Suzanne Filteau[1]

¹Department of Population Health, Faculty of Epidemiology and Population Health, London School of Hygiene and Tropical Medicine, London, UK
²School of Public Health, University of Zambia, Lusaka, Zambia
³ZAMBART, University of Zambia School of Medicine, Lusaka, Zambia

**Correspondence to**
Dr Sehlulekile Gumede-Moyo;
sehlulekile.gumede@lshtm.ac.uk

## ABSTRACT

**Objective** This study aimed to investigate the challenges in implementing a Zambian electronic health records (EHR) system labelled 'SmartCare' from diverse stakeholder perspectives in order to improve prevention of mother-to-child transmission (PMTCT) data collection so that SmartCare can be used for clinic performance strengthening and programme monitoring.

**Design** This is a qualitative retrospective study.

**Setting and participants** SmartCare is a Zambian Ministry of Health (MoH)-led project funded by the US Centre for Disease Control and Prevention. Data were collected using in-depth interviews, observations and focus group discussions (FGDs) between September and November 2016. Seventeen in-depth interviews were held with a range of key informants from the MoH and local and international organisations implementing SmartCare. Four data entry observations and three FGDs with 22 pregnant and lactating women seeking PMTCT services were conducted. Data were analysed using a thematic content approach.

**Results** The SmartCare system has evolved from various patient tracking systems into a multifunctional system. There is a burden of information required so that sometimes not all is collected and entered into the database, resulting in poor data quality. Funding challenges impede data collection due to manpower constraints and shortages of supplies. Challenges associated with data collection depend on whether a paper-based or computer-based system is used. There is no uniformity in the data quality verification and submission strategies employed by various IPs. There is little feedback from the EHR system at health facility level, which has led to disengagement as stakeholders do not see the importance of the system.

**Conclusion** SmartCare has structural challenges which can be traced from its development. Funding gaps have resulted in staffing and data collection disparities within IPs. The lack of feedback from the system has also led to complacency at the operational level, which has resulted in poor data quality in later years.

## INTRODUCTION

Routinely collected clinical data can provide much needed information on the prevalence

## Strengths and limitations of this study

► We obtained a rich and nuanced appreciation of the implementation of the SmartCare as we sampled from a range of implementing partners, health facilities, stakeholders and implementation systems.

► The depth of the inquiry enabled us to consider a range of explanatory factors.

► Data collection was restricted to health facilities in and around Lusaka, the capital city of Zambia, and therefore, the health facilities selected could be receiving financial and technical support of an order that is unlikely to be offered in remote facilities.

of HIV among pregnant women and the uptake of services for prevention of mother-to-child transmission (PMTCT) of HIV.[1] The use of routinely collected data can be timely and cost-efficient for decision-making as data are already available for analysis.[2] Collection of high-quality routine data on these services and outcomes for HIV-positive mothers and HIV-exposed infants is essential for monitoring and evaluation of PMTCT programme, for clinical management of patients and for managing stocks of HIV test kits and drugs.[1] For both clinic staff and health system managers, having access to reliable data that reflect the processes of care and clinical outcomes is the first step to ensuring effective delivery of an intervention within a healthcare system.[3 4] However, in Zambia there is underutilisation of routinely collected data in the HIV programme.[5]

Data collection systems for PMTCT programme in Africa often lack immediacy as many are paper-based with records completed at each facility, and with individual level data that are inaccessible to central planners.[6] Electronic health records (EHRs)

could present an opportunity to supplement current sources of routinely collected surveillance data.[7] EHRs are real-time, patient-centred records that make information available instantly and securely to authorised users.[8] While an EHR does contain the medical and treatment histories of patients, an EHR system is built to go beyond standard clinical data collected in a provider's office and can include a broader view of a patient's care.[9]

Various EHR systems are implemented globally.[10–12] A systematic of review of literature about EHR systems in resource-constrained settings recommended that it is urgent to evaluate barriers to implementation.[13] Our own research on extracting surveillance data from a Zambian EHR system, SmartCare, has highlighted some deficiencies including large amounts of missing data, especially in more recent years, and variable performance across the country.[14 15] This study aimed to investigate the challenges in implementing SmartCare from diverse stakeholder perspectives in order to improve PMTCT data collection so that it can be used for clinic performance strengthening and programme monitoring. For the context of our study, implementation of SmartCare was based on getting good data from individuals attending health facilities. The later stages involving retraining of staff and reorganising supply chains are not included in this definition of implementation.

## METHODS
### Study design
This qualitative retrospective study included in-depth interviews, observations and focus group discussions (FGDs) conducted between September and November 2016.

### Study setting
SmartCare is a Zambian Ministry of Health (MoH)-led project funded by the US Centre for Disease Control and Prevention (CDC). SmartCare was developed to improve continuity of care and provide timely data on maternal and child health, HIV/AIDS, tuberculosis and malaria interventions for public health purposes, trend reporting and analysis for health officials and clinicians.[16] The implementing partners (IPs) are responsible for data entry mainly from paper-based forms which have information collected by clinicians, which is then entered into a computer.

The SmartCare database is a derivative of the PTS (patient tracking system), which was developed in 2004, based on a health facility-centred EHR system. In 2010, it became a national medical health programme and was then rolled out throughout the country. It is implemented by government, and both international and local organisations primarily for patient management. Most of the international organisations were involved in the database development, and the local organisations became involved when implementation was rolled out throughout the country (after 2009).

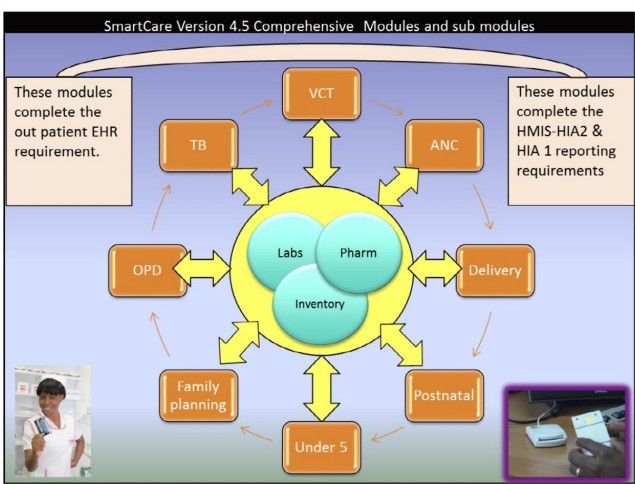

Adapted from Ministry Health HMIS workshop Presentation 2016 - *Permission granted from the copyright holder*

**Figure 1** The Ministry of Health of Republic of Zambia is the copyright holder of the figure. ANC, antenatal care; EHR, electronic health records; HMIS, health management information system; HIA, health information aggregation; OPD, outpatient department; TB, tuberculosis.

SmartCare is organised into comprehensive modules and submodules (figure 1). This was mainly influenced by its funders depending on the information they wanted at that particular time. The main module groups are clinical group for which the modules of interest in this study are Antenatal care (ANC), Delivery, Postnatal, Antiretroviral therapy (ART), Paediatric ART, PMTCT follow-up, and Under 5; logistic group which has information on drug dispensing and orders; monitoring and evaluation groups which includes health management information system reports, graphing, data analysis, data merge from facilities for MoH; and the continuity of care group which has data from across facilities and within facilities. Through the data merge, the SmartCare information can be used to obtain data for the monthly reports to the MoH.

Data are captured under a number of 'modules' across a range of health issues. Records are updated at every point of clinical service. Patients are issued with Smartcards at their initial consultation which contains all their clinical information and treatment details and can be accessed from any SmartCare facility.

The SmartCare data are collected either directly onto a computer or using a paper-based method. When using the computer-based data collection and entry method, clinicians have to enter data directly in the computer; this method is still being piloted in a few facilities mainly in the Southern and Lusaka provinces. For the more common paper-based data method, client information is documented in the SmartCare care forms. The papers are then taken to the data entry room where the information is entered into the computer. Some facilities which are deep rural (where there is no electricity at all) use paper

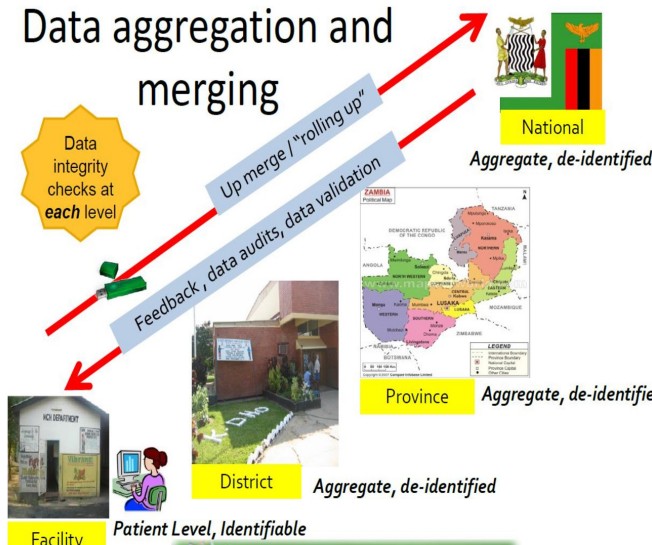

Adapted from Ministry Health HMIS workshop Presentation 2016 - *Permission granted from the copyright holder*

**Figure 2** The Ministry of Health of Republic of Zambia is the copyright holder of the figure.

only and the forms are then transferred to a location where there are computers and entered into the database.

Data on a range of HIV and pregnancy-related outcomes are collected during the patient consultation with the clinician, who records the information on paper forms or directly on the computer in the few facilities which are computer-based. The paper forms are then entered into the SmartCare database by data entry clerks who have been trained in using SmartCare. Data are collected from each facility on a monthly basis and submitted to a district health information officer, who aggregates and sends the data to the province level senior health information officers. From here, the data are wired to a national server at MoH headquarters (figure 2).

For the purposes of this study, routine data are defined as data that are routinely generated through ANC and PMTCT service delivery, and routinely recorded in standard SmartCare data tools.

## Sampling

Data were generated from 17 in-depth semistructured face-to-face interviews, 4 data entry observations and 3 FGDs with 22 pregnant and lactating women seeking PMTCT services from 3 health facilities (table 1).

## Key informant interviews

The IPs were purposively sampled in consultation with the MoH to identify diverse organisations. Six health facilities were selected based on local considerations which included both well-resourced international organisations and less well-resourced local and government-owned organisations; those best performing versus struggling; those which are paper-based or computer-based; and facilities which had already transitioned to 'test and treat' for general population HIV care. Most of the facilities sampled were in Lusaka City, and only one was peri-urban. The choice of Lusaka was based on the following considerations: (1) logistics, (2) facilities in Lusaka are busy and have the highest case loads, (3) our previous work has shown that facilities in Lusaka show a wide range of successes and failures, and (4) overall Lusaka performs poorly compared with the rest of the country and therefore it was important to investigate the working of Smart-Care in Lusaka City.

Within each of the selected IPs, a diverse range of interviewees were purposively recruited in order to give the broadest range of perspectives.[17 18] The approaches to recruitment of participants were flexible, being negotiated with key contacts and recommendations from managers who were also considered for in-depth interviews. Interviews with key informants were conducted by the first author in English. They lasted 30–60 min and consisted of a series of open-ended questions with follow-up question probes. The interview guides were specific to the hierarchical levels of operation, namely leadership level (persons involved in the strategic direction of SmartCare); operational level (data managers and monitoring and evaluation personnel); and implementation level (data entry clerks and MCH in-charge clinicians). The interview guides were designed to capture

**Table 1** Breakdown of in-depth interviews and focus group discussions participants

| In-depth interviews with key informants | |
| --- | --- |
| Leadership | 2 implementing partners, 1 Ministry of Health representative |
| Database management | 1 Ministry of Health ICT officer, 2 Data managers, 2 Monitoring and evaluation officers, 1 database developer |
| Data entry clerks | 1 computer-based health facility, 4 paper-based health facilities |
| MCH nurse-in-charge | 1 computer-based health facility, 2 paper-based health facility |
| **Focus group discussions with service users** | |
| Peri-urban health facility | 4 pregnant, 2 lactating women |
| Paper-based health facility | 4 pregnant, 4 lactating women |
| Computer-based health facility | 4 pregnant, 4 lactating women |

ICT - Information and Communications Technology; MCH Martenal and Child Health

information on SmartCare development, implementation procedures, data quality management and linkage between SmartCare and PMTCT services.

## Focus group discussions

The FGDs were conducted at maternal and child health (MCH) departments with participants from three different facilities, two from Lusaka and one from the peri-urban area. The participants were recruited with the assistance of PMTCT coordinators who were instructed to balance the number of pregnant and lactating women on ART. The FGD sessions were conducted by the first author and a co-moderator who was fluent in the local languages (Bemba and Nyanja). These discussions were guided by an interview guide containing open-ended questions and lasted for 60–90 min. The interview guide was designed to capture the general knowledge and experience of the PMTCT programme and SmartCare operations.

## Facility observations

The observations were conducted to familiarise the researcher with the realities of the SmartCare data collection and entry environments. These were planned to be conducted in six health facilities through interaction with study participants and relevant actors, as well as personal reactions to the related events without the use of a specific tool. However, in two health facilities, data entry was not taking place on the scheduled visit days due to power outage; hence, only in-depth interviews were conducted with implementing level staff.

## Data management and analysis

The in-depth interviews and FGDs were tape-recorded and subsequently transcribed verbatim. FGDs were then translated into English. Data were collected aiming to get sufficient data from relevant actors/perspectives to ensure it was qualitatively representative.[19] Detailed field notes were taken on the sites following the observations for analysis.

A thematic content approach was conducted; this was done by reading the transcribed material systematically, line by line, in order to identify the meaning units. Meaning units were defined as strings of the text expressing a single coherent thought, up to the point at which the coherent thought changed. Thereafter, the meaning units were marked by a code, describing what the text unit was about. Coding was also done on issues that were related to the reasons for missing PMTCT data; text unrelated to these issues was not included in the present analysis.

The codes were defined and organised so that those referring to the same subject were grouped into categories. The interview guide was used as a starting point for grouping the information, but during the analysis new categories were developed. When this happened, we re-coded. The data were managed manually in Excel sheets. The underlying meanings of the categories were formulated into a theme. The original data were re-assessed by an assistant (MSc student) after analysis in order to detect any concepts or information that would have been missed. The first author discussed the results with SF, JT and the MSc student.

## Research ethics

Written informed consent was obtained from all key informants including their consent to record the interviews and publish anonymous quotations. Verbal consent was sought for FGD participants. Participation in the study was entirely voluntary and participant anonymity was maintained throughout the processes of interview transcription, data analysis and presentation by using pseudonyms. Permission to collect data from the health facilities was granted by the MoH district offices.

## Patient and public involvement

Patients were not involved in the development of the research question, design of the study and conduct of the study. The findings will be shared with the policy makers in order to help them to improve the implementation of the EHR system.

## RESULTS

The comments from key informants, observations and the FGDs enabled us to characterise four fundamental issues that are key in the implementation of SmartCare: database development and ownership, funding and staffing, the data collection process and health facility setup.

## Database development and ownership

The development of SmartCare was mainly CDC-led, with the government of Zambia through the MoH supporting the process. Multiple stakeholders using various patient tracking software were brought together to form SmartCare. Reflecting on the history of SmartCare, two IP informants identified the leading role of CDC and the subsequent collaboration with a broader group of partners.

> The initial development was by the American-funded institutions and a few people from the MoH because they had gone into partnership. They were developers at MoH and also at CDC, the rest of the institutions like us were only required to use the software.—Implementing partner leader

> There were other systems that were used by other partners throughout the country for example, there was a system called the CareWare which was similar to SmartCare but it was just a patient summary. Then there was another system which was used in the private sector which l can't remember the name. Lessons were drawn from all these systems and brought into one and this is how SmartCare was born.— Implementing partner leader

> There was a consensus among the leaders of the SmartCare implementation that the database has also evolved from a system to track patients to its current form as a

mixture of a patient tracking tool, a clinical care tool, a reporting tool and a surveillance tool.

> …it was just a mixture of ideas which were clinically required but each person had different needs for information, some needed it for research purposes, 1 needed it for patient follow-up, another person needed it for forecasting and quantification.—Implementing partner leader

The different stakeholders had different data needs which took the form of added modules. The changes have, however, made it difficult to classify and use the database because it now tries to address multiple issues simultaneously.

> I wanted my things as well and partly 1 wanted things that PEPFAR (The United States President's Emergency Plan for AIDS Relief) wanted because our money comes from them. If I can't report on PEPFAR indicators, 1 don't get more money.—Implementing partner leader

The resulting SmartCare forms are 6–7 pages for a normal interaction per visit because of requirements from its various stakeholders. In addition, a lot of data have been entered into the system, making it bulky. This has caused the system to become very slow, and in the mornings it takes a very long time to reboot as narrated by operational and implementing level participants.

> Because the system is slow they have to wait instead of entering 100 files per day they are only able to enter 60 or so.—Database manager

Data entry is often interrupted as the system also hangs up every now then because it is so bulky. Hence, a lot of backlogs are experienced, which results in a lot of data not being entered.

> Last year it had serious faults and we were not using it for a while.—Data entry clerk

Although there is a SmartCare technical working group, chaired by the MoH where operations are discussed, it is not clear who is guiding the process, and how the database should be modified to make it work better.

### Funding and staffing
The database operations are managed by the monitoring and evaluation (M&E) departments of the IPs. The structures of the M&E departments differ across the IPs, with international organisations having much bigger and better resourced departments. The local IPs have much leaner structure, rely mostly on volunteers and run into problems due to financial constraints. This was elaborated by the leadership and implementing level participants.

> Mostly the people who enter data in the health facilities where we operate are employed by us. Apart from that we also provide support through volunteers; in

most cases we also provide peers and counsellors.—Database manager

The manpower constraint in the local partners has resulted in the loss of data during long grant negotiations. Data are lost as data entry staff are laid off between funding cycles since their salaries are dependent on grants. In addition, these partners have designated personnel who enter general ART data, while PMTCT data are mainly entered when the data entry personnel have specific interest or have trained temporary volunteer staff.

> The positions of data entry clerks in the facilities are actually paid for by the donor. It's not sustainable because the moment the donor says we are pulling out, there will not be any data entry personnel in these facilities.—Implementing partner leader

The funding of operations also affects the supplies required for the day-to-day operations as narrated by both operations and implementing personnel.

> We normally run out of forms, especially when funding is out. When we don't have funding the facilities also have challenges in printing the forms because toner is expensive.—Data entry clerk

> …when the computers are down because of the virus, data entry doesn't happen.—M&E officer

In some small facilities where the MoH is solely responsible for data collection and entry, there are no data entry clerks and hence clinicians enter the data on the computer after hours. This is usually done only by staff that are passionate and so give it time according to an MoH leader.

> In the government facilities there is no designated person to manage ART data for SmartCare; its health workers who just do that on their own.—Implementing partner leader

### Data collection process
#### Data entry
With both the computer-based and the paper-based methods, the greatest challenge is incomplete forms which translate to missing data in the database. In some facilities which are computer-based, the data entry clerks noticed that there are also some clinicians who still prefer entering data on paper and handing over to the data entry clerks to enter on the computer.

> In as much as data are supposed to be entered at each service point, some people prefer entering the data on paper and they hand over the data to me.—Data entry clerk

According to implementing level staff, with the paper-based method, once the client leaves the facility it is a challenge to follow-up on the missing data on the forms.

> There are usually many gaps in the forms; the forms are not fully completed more than half of the time.—Data entry clerk

When the forms are not complete l return them to whoever was supposed to be entering the data at that service point but the challenge is that they will tell you the person has gone and that can't follow them up and even so in the next visit they will not even bother correcting the information.—Data entry clerk

One of the MoH participants explained that the incomplete data forms could be a result of the competing demands to physically attend to clients and fill out the forms in the high volume facilities.

… It takes 20–30 min just to enter the data for one interaction. In clinics where there are many patients, how many patients are you going to see in a day?—Implementing partner leader

The implementation challenges of the SmartCare database depend on the data collection and entry method used. Power outages were the greatest challenge: under the computer-based method, client information is not collected, as some clinicians will be reluctant to use the alternative method, whereas with the paper-based method there will be a backlog of work.

Some well-funded partners have installed solar power systems in the health facilities, and this enables them to enter data without interruptions. Despite having good power supply, there is a challenge in these facilities of some clinicians who are not computer literate. Furthermore, because of other competing health facility priorities, the power is rarely used for data entry.

…would you want to power theatre activities or you want to power computers for data entry? Theatre is life saving and data can be entered anytime.—Implementing partner leader

### Data quality verification

Verification of data quality is specific to each IP. The verification process involves comparing the patient information on the paper forms and registers against the information in the database. The international partners also have verification procedures imbedded in the monitoring and evaluation strategies as narrated by one of their participants.

We are doing double data entry for 10% of our files.—Data manager

On the other hand, verification is rarely done by the local organisation due to manpower constraints as indicated by an operational level participant.

When l get time, l check what is in the system vs what is in the registers. However it's rare that l get that time because l always have files to enter into the system. In that case l wait for the Quality Control people from Head Office to come and do random verifications for me.—Data entry clerk

All the partners reported that there are some instances where they cannot find patient files, making it difficult

for them to verify the data that is in the computer with the raw data.

There are certain instances where we can't find patient files.—M&E officer

The patient files will tell me that a facility has 200 clients whilst SmartCare has 300 clients; this is because some of these files are kept somewhere, could be files of staff members or people with prominent positions in that particular area.—Database manager

The SmartCare Database system has reports that are built in the application. However, all the partners alluded that these reports are very inconsistent and incorrect.

In some cases it will show a large number of people who would have been initiated in a day more than what we would have actually done.—MCH nurse-in-charge

### Submission of data to the main database

There is also no uniformity in submission of the data to the main database, although according to the MoH the data are supposed to flow from the facility to the district, province and then to the main database at head office (figure 2).

They sometimes come to the facilities and collect the data on their own since they are the ones who own these facilities as well.—M&E officer

Some partners wait for MoH to request data while some submit their data to CDC.

We do not follow that procedure as per say because we collect data directly from the sites and aggregate it from here and submit to CDC. If MOH requests it, we submit to them but we are not that consistent.—Database manager

### Feedback from database

There is no standardised procedure for reporting data that are collected by IPs. The partners normally use the information internally as well as sharing with the donors supporting them.

We also have special reports that we do in a monthly basis such as the PEPFAR report.—M&E coordinator

In contrast, the nurses-in-charge in some of the facilities reported that they have never received feedback from the system.

We have not received anything; they don't give us any reports. We would, however, want to receive reports so that we would know how we are faring.—MCH nurse-in-charge

As viewed by one of the leaders who participated in the study, the lack of feedback from the system has led to disengagement and some stakeholders not seeing the importance of documenting client information in the SmartCare database.

A lot of the staff collecting this data do not understand how and why they are collecting this data and they are frustrated by it as it also takes lots of their time entering this data.—Implementing partner leader

## Health facility setup

The setup in some health facilities is in such a way that the PMTCT services are offered in the ANC department, whereas ART services are offered in places designated solely for ART. The nurses-in-charge in such health facilities narrated that the facility setup could be a great challenge in ensuring that the PMTCT data are entered in an efficient manner.

Probably if it was housed within the ART department, you could have the same person entering the PMTCT data into the SmartCare.—MCH nurse-in-charge

In addition, the MCH nurses-in-charge indicated that client files sometimes go missing between the MCH and the data room.

I am sure when they are sending the files; some are lost during transfer in different departments.—MCH nurse-in-charge

## Views of pregnant and lactating women

Overall, women seeking PMTCT services did not have concerns on the implementation of SmartCare, as according to them the healthcare workers will be doing their job.

The person who is supposed to enter the data does so because he was trained anyway.—PMTCT FGD mothers

We just carry our cards and leave them to do their job.—PPMTCT FGD mothers

We are here for our lives and that of our babies, we have to comply and be patient.—PPMTCT FGD mothers

## DISCUSSION

The study has provided evidence on the SmartCare implementation challenges. It has also provided insights as to how to improve implementation and data quality. The main points were to collect less, but better data, to engage the clinic staff by providing regular feedback and to improve the software. Fortunately, PMTCT clients appear already satisfied with the system.

The design of EHR systems does not always get attention due to pressures related to their functionality requirements.[10] SmartCare was exposed to pressures as its data model evolved from a patient tracking system into a multifunctional system due to demands of various stakeholders. In an effort to shape the SmartCare database into a manageable tool that can be used for public health purposes, CDC and MoH could streamline the process for including and excluding information that is collected. There is a need to cut down on the number of fields collected so that less data of higher quality are collected. This could improve the system by reducing the amount of data so that it takes less time to collect and people might more readily do it; and having a less bulky database which will be less likely to be affected by computer crashes and faster to use.

SmartCare could be developed into a networked EHR, which has been reported to be successful in other developing countries like Rwanda and Kenya.[20] In a networked EHR, data can be accessible and be shared in multiple sites; multiple users can enter data simultaneously; data can be backed up automatically at more than one site; information can be communicated between multiple locations; data are accessible and shared at multiple sites and the system can be debugged and upgraded over the internet without visiting remote sites.[10 21] This would allow greater use of the data by health workers and Ministry planners. However, it is important that there is uniformity in the implementation of SmartCare throughout the country so that everyone will be on the same level. This will allow for comparisons across geographical regions and longitudinal analyses.

In Ethiopia, the quality of data was reported to be affected by dual documentation where both paper-based and electronic systems were used.[22] In this study, completeness of paper-based records was slightly better than electronic records. According to Odekunle *et al*, limited computer skills is one of the factors affecting electronic health record adoption in sub-Saharan Africa. The authors pointed out that in some countries, many physicians and other key end users are not eager to adopt an EHR, resulting in low EHR adoption in the region.[23]

There was a notable lack of appreciation of the system, and a need to train and support end users of the system such as the clinicians who are directly involved in the data collection process. For both the clinic staff and health system managers, having access to reliable data that reflect the processes of care and clinical outcomes is the first step to ensuring effective delivery of an intervention within a healthcare system.[4] Hence, there should be better buy-in of the staff at the facilities in order to make sure that they fill in the forms regularly and accurately.

In our findings, workload affected the documentation, as evidenced by poor data quality in the quantitative analysis of SmartCare PMTCT data.[14 15] In low-resourced countries there is a shortage of a qualified workforce and most healthcare institutions do not have dedicated IT staff for their EHR systems.[24] The results of our study are consistent with a systematic review of literature on the role of EHR systems in developing countries, which pointed to the lack of financial and human resources as major challenges in implementing EHR systems.[20] Therefore, there is a great need for strong commitment from both the government and the donors to invest in improving the implementation of the system.

The data from the database should be presented to facility staff as an advocacy tool which is needed in their facilities, so that they would appreciate the need for it.[21 25] Similarly, in their description of the rationale and experience in scaling up EHR in Malawi, Douglas and colleagues concluded that health workers will not adopt a system if they do not find sufficient value in it.[26] It is also possible that health workers perceive SmartCare as complex and of little value to their work; therefore, any revision to the system would require a clear strategy for ensuring that the operations are well conceptualised.[27]

The knowledge required for the use of the database should easily be transferred through training, ensuring that it is customised to the IPs' requirements and the establishment of helpdesk call centres to readily assist with operational challenges. There is also a need to strengthen interorganisational networks among IPs by promoting collaboration aimed at sharing ideas and knowledge construction.

### Strengths and limitations of the study

We obtained a rich and nuanced appreciation of the implementation of the SmartCare as we sampled from a range of IPs, health facilities, stakeholders and implementation systems. The depth of the inquiry enabled us to consider a range of explanatory factors. Most of these factors were related to the EHR system and its implementation. We also considered the factors encountered by the PMTCT clients but these did not emerge as major concerns. This is likely due to the fact that the patients do not interact with SmartCare directly.

There are several limitations of the study, such as the restriction to health facilities in and around Lusaka. Lusaka is the capital city of Zambia and therefore the health facilities selected could be receiving financial and technical support of an order that is unlikely to be offered in remote facilities even by the same IP. However, our previous work on the quantitative analysis of PMTCT data does not support this argument as several provinces had better data quality than Lusaka.[14 15]

Another limitation of the study it is that, although we intended to focus on implementation of SmartCare for PMTCT, the issues that arose were related to the entire system. The findings of the study might have been different if stakeholders from other health services such as general ART services were included. The system in Zambia is such that pregnant and lactating women are given 'VIP' treatment, which allows them to be served faster under the ANC department compared with those under general ART. Despite that the setup offers 'VIP' services, data collection and entry of PMTCT services were noted to be a challenge as in most facilities there is no designated personnel for PMTCT data entry. In addition, the movement of forms from the PMTCT department to the data entry room was reported to be not well coordinated.

Our study did not investigate how patients relate with the clinicians when they interact with them; although this could have had an impact on both staff and patients' perceptions of SmartCare, it was beyond the scope of our research.

### CONCLUSION

The SmartCare system has structural challenges which can be traced from its development. Funding gaps have resulted in staffing and data collection disparities within IPs. The lack of feedback from the system has also led to complacency in operational level, which has resulted in poor data quality in later years. The data from the database, if appropriately understood, could be used by health facility staff as an advocacy tool, as well as in monitoring the impact of PMTCT programme. Our research could aid other countries wanting to develop their own EHR systems.

**Acknowledgements** We also thank the Ministry of Health, Department of Policy and Planning for granting us permission to access the SmartCare database. The authors thank all the participants who contributed their time and effort to the study and Dr Jasleen Singh (MSc student) for going through the transcripts and assessing the initial analysis.

**Contributors** SGM conceived the study and conducted all the interviews and observations under the supervision of SF and JT. Analysis was conducted by SGM with the input from SF, JT, PM and VD. The manuscript was drafted by SGM with contributions from all the authors. All authors read and approved the final manuscript.

**Funding** The study was supported by the SEARCH (Sustainable Evaluation through Analysis of Routinely Collected HIV data) Project funded by the Bill & Melinda Gates Foundation grant number OPP1084472.

**Competing interests** None declared.

**Patient consent for publication** Not required.

**Ethics approval** The study was approved by the Zambia Biomedical Ethical Board (Ref 101-04-16) and the LSHTM Research Ethics Committee (Ref 12086).

**Provenance and peer review** Not commissioned; externally peer reviewed.

**Data availability statement** No data are available.

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
