## [Reviewer comments · BMJ Open]

ARTICLE DETAILS

TITLE (PROVISIONAL)	A qualitative inquiry into implementing an electronic health record system (SmartCare) for prevention of mother-to-child transmission data in Zambia- A retrospective study.
AUTHORS	Gumede-Moyo, Sehlulekile; Todd, Jim; Bond, Virginia; Mee, Paul; Filteau, Suzanne

VERSION 1 – REVIEW

REVIEWER	Nancy Puttkammer University of Washington
REVIEW RETURNED	23-Apr-2019

GENERAL COMMENTS	Summary This paper presents results of a qualitative study on implementation of the SmartCare system for PMTCT programs in Zambia, based on 17 in-depth interviews with system stakeholders, 3 focus groups with program clients, and observations of system use. Overall comments: This paper is an important qualitative case study of the Zambia SmartCare system. There are relatively few in-depth qualitative studies of health information systems in low-resource settings, so this study is poised to make a contribution to the literature. The results presented are important and the overall conclusions seem very credible. The strongest gaps in the paper are: 1) the lack of a clear research question; 2) limited coherence in the justification of the sampling strategy; and 3) the lack of conceptual or theoretical grounding of the study analysis and results. Overall, stronger contextualizing of the study findings in the Discussion section and stronger writing could help push this manuscript to make a greater contribution to the literature. SmartCare seems to represent a “first generation” electronic health record in the Zambian setting, with some of the same weaknesses that have hampered other similar systems. Greater linkage of the study findings to the literature on best practices in EMR design and implementation would be good. Major comments • The authors state in the Introduction “The overall goal of our study is to improve the implementation of an EHR system in Zambia labelled ‘SmartCare’ and address the fundamental need for timely and high-quality data” (page 4, lines 50-54). This explains the motivation for the study, but not the central research question. This ambiguity is a problem throughout the manuscript. The study design and sampling strategy was also not clearly tied to a research question. Their purposeful sampling scheme lacked
--

	some coherence because they indicated that they sought sites with both stronger and weaker data quality. This suggests a research purpose to identify factors contributing to strong data quality in some sites and weak data quality in others. However, they also indicate that they included only Lusaka where data quality tended to be weaker, suggesting that they did not actually try to contrast factors in stronger vs. weaker sites, but rather sought to identify challenges in weaker sites only. The fact that they interviewed a range of stakeholders including clients suggests that perhaps their research purpose was to describe perceptions of the success of implementation of SmartCare from diverse stakeholder perspectives. In any case, a clearer statement of the research question would be helpful. The abstract includes a somewhat clearer statement of the study aim: “This study aimed to investigate the challenges in implementing a Zambian EHR system labelled ‘SmartCare’ in order to improve PMTCT data.”  • Discussion page 17, lines 15-17: The authors state: “The depth of the inquiry enabled us to consider a range of explanatory factors. Most of these factors were related to the EHR system and its implementation.” However, their study lacks a clear research question -- they do not identify what they are trying to explain (explanatory factors for what?). • Methods: the authors should discuss how the 22 participants of the FGD were recruited, and any incentives offered. The authors should describe the setting from which the participants arose (if the FGD participants were recruited from 3 different clinics in Lusaka, for example). • Conceptual basis of the study. While it is certainly the authors’ right to choose to come from a theory-free perspective in designing and carrying out the study (“Data were collected and analysed without a constraining overarching framework, which enabled us to capture the diversity of experiences of the SmartCare implementation efforts”), I found the lack of a conceptual basis to be a weakness in the presentation of the Results and the Discussion. In particular, I felt that in the Discussion section, the authors could have done a better job presenting and discussing their findings in reference to the broader literature. Examples of papers which might be of use in shaping the structure of the Discussion section include:  o Fritz F, Tilahun B, Dugas M. Success criteria for electronic medical record implementations in low-resource settings: a systematic review. Journal of the American Medical Informatics Association. 2015;22(2):479-488. o Khoja S, Durrani H, Scott RE, Sajwani A, Piryani U. Conceptual framework for development of comprehensive e-health evaluation tool. Telemed J E Health. Jan 2013;19(1):48-53. Following a conceptual framework or set of categories like that identified by Fritz or the KDS paper, might help in organization of the Discussion section. • Results: The client perspective is described only very thinly in the results (page 15, lines 25-33). This does not offer much insight on the patient perspective. I encourage the authors to see if there is any additional meaning or insight to be gained from the client FGDs. • The recommendation on upgrading to a networked EMR (final paragraph of Discussion section) reflects a mismatch in understanding about the OpenMRS platform (see https://openmrs.org/) and about other EMRs deployed in low resource settings. Is the recommendation to convert the system to the OpenMRS data model and platform? Or to enhance
--	---

	SmartCare to encompass some of the features mentioned in the paragraph? It is not clear that the features mentioned would address some of the main implementation challenges the author has identified (too much data being collected, without value added for front-line personnel; mixed technical performance; lack of staffing; lack of feedback and data use).  • Limitations: A limitation of the study seems to be that it focused on implementation of Smartcare for PMTCT, but addressed issues which seem to relate to the entire system. Do the authors think that their findings would have been similar or different, had they also included stakeholders (health workers and clients) from other health services like TB or HIV ART services? Minor comments:  • Introduction. The authors indicate they completed a systematic review of literature (page 4, line 25). Usually a systematic review is an end in and of itself, and involves citing the databases searched, the search terms used, and how the review was done. It's not clear to me that the authors followed a formal systematic review process. It might be best to drop the word "systematic". • Sampling: before diving into the discussion of sampling on page 6, line 37, it would be best to reiterate the types of qualitative data collection and number of interviews done (i.e. move text from page 7, lines 16-19 to start the sampling section). • Please copy edit throughout. Please fix typos and grammar problems (e.g. page 7, line 30 refers to "stuff" rather than "staff"). • Need to explain the role of IPs in Zambia's health system, when they are introduced in the manuscript. A reader without knowledge of PEPFAR-supported programs may not understand the phrase "IPs". • The hybrid data collection process described on page 11, line 50 – page 12, line 5 should be moved to the Methods section where the system itself is described. • Figure 1: Authors should define abbreviations, like "HIA 1", "OPD" etc. • Discussion page 16, line 47: "There was a notable lack of appreciation of the system". The authors should clarify who had a lack of appreciation of the system. The authors could underscore the diverse purposes of the system as it was originally conceived and the lack of user-centered design processes during software development. • The authors present the following result without context, and fail to return to it in the Discussion section (page 13, lines 50-51): "The SmartCare Database system has reports that are built in the application. However all the partners alluded that these reports are very inconsistent and incorrect." Is this simply a perception of end users, which is explained by poor data quality leading to reports that do not seem to reflect reality? Or is it an actual problem of faulty queries producing the automated reports, which relates to issues about lack of financial resources for sufficient design and testing of in-built reports in the system? Or do the authors have limited information to "unpack" this perception from end users. Giving some further indication of the nature of the problem, or at least discussing it further would be helpful.
--	---

REVIEWER	Abigail Baim-Lance City University of New York Graduate School of Public Health and Health Policy
REVIEW RETURNED	29-Apr-2019

GENERAL COMMENTS	Abstract The abstract results do not highlight the same issues as the results section in the paper. A major issue highlighted in the paper is the burden of information needing to be entered, which is not mentioned at all in the abstract. It might make sense to use the four (or three – see below) major findings issues listed in the paper to structure the abstract results section. Introduction This paper begins by describing the benefits and barriers to using routinely collected EHR data for PMTCT surveillance purposes, with a focus on Zambia which uses the SmartCare EHR System. Barriers to use of SmartCare have been identified through a literature review and preliminary work, which focuses mainly on the quality of the EHR data, such as missing data components. The stated goal of the study is then to build on these reviews to “improve the implementation of the EHR system in Zambia and address the fundamental need for timely and high-quality data” (pg 4). To do this, the authors employ a qualitative approach of focus groups and interviews to presumably understand the issues related to use of the EHR data system. There are primary two concerns here. The first is that the introduction does not state the investigative purpose of the study; one imagines that system improvement would be an outcome from some kind of empirical study, but of what? The abstract states this more clearly: “the study aimed to investigate the challenges in implementing...” This or similar explicit language about aims needs to be brought into the introduction to state the purpose of the study at present, not only its implications for future. A second concern revolves around the degree to which the study can illuminate issues related to use of SmartCare for PMTCT monitoring purposes: likely a more important outcome of a study such as this, but one that is different from the focus on the SmartCare EHR system in general. The authors do not distinguish these different types or steps in implementation, and I would suggest attending to them each by clarifying what is meant in this paper by ‘implementation’ (as well as what questions cannot be answered by this study). They can then address this again by discussing the implications of results in the discussion and limitations sections – see some additional comments about this below. Methodology Sampling “The approaches to recruitment of participants were flexible” – what does this mean? What was the negotiation? It is unclear how the recruitment actually occurred. How were the patients recruited, all told 22? That’s a big number; how were they selected, and how was representative ensured? Instead of figure 2, it would be helpful to include a summary table that breaks down the types of interviews by categories included on bottom pg 6, para beginning line 54. High/low performing; paper/computer. Later when citing quotations, it would be helpful to refer to the interviewee by clinic type Analysis
--

	The grounded description of analysis of ‘no overarching frameworks’ or ‘pre-conceived ideas’ does not seem plausible. Beyond that there is generally a strong applied steer in this kind of work, the authors state the goals of the study to explain specific “deficiencies” in optimizing the SmartCare system, so it is much more reasonable to assume that these areas guided initial as well as later coding. If a different ‘unconstrained’ approach was used, then cite to the method, justify and describe it more persuasively. Results The four primary thematic areas/issues seem fine, though would suggest numbering them in the first results paragraph for clarity. When describing each issue, use of quotes needs better attention. A quotation needs to be set up, including part of a sentence rather than its own sentence (such as, “As X described, “.”) and each quotation needs to be contextualized or explained. This may happen prior to a quote, but it often happens after. If a second quotation is used, it should present another idea or a nuance or elaboration of the first, and again the authors need to point this out for the reader rather than simply including a second excerpt. At present, sometimes two quotations side by side don’t seem to be making the same point (e.g., top of page 10; how does PEPFAR performance relate to the range of clinical inputs?). At other times, the prose is unsupported by the text (bottom of pg 10). I advise authors to make each selected quotation more robust by explaining how to understand them, and giving more context to interpret them. Again, a summary table can then be used to show how each interviewee is located across salient categories. The ‘facility set up’ issue 4 section is very thin, and perhaps can be incorporated into the 3rd issue, which seems very process oriented. The quotations from the focus groups are also quite thin; if the patient interviews are not useful to this analysis, they should be removed. At present, they feel tokenistic, which is strange if 2 were completed with over 20 individuals so to sum up their contribution with 3 quotes that have nothing to do with data does nothing for the argument. It is understandable that patients may not have awareness of the more technical issues of data entry and use, so pulling the patient piece out may make sense. Discussion Consider citing to implementation science literature (e.g., diffusion of innovations – Greenhalgh et al 2004) about how organizations adopt new innovations, or new practices. It is gestured to in feasibility, but the authors could bring in other categories as well. When describing the importance of gaining buy-in among end users at bottom of pg 16, the discussion begs the question of how this is achieved, and turning to implementation science lit might help to offer some suggestions. Again, it would be great to talk about implications of findings not only for getting better data into the system, but using it for program monitoring – not the same activity! Limitations Though the paragraph states several limitations, only 1 is listed. What are the others? One obvious issue (again) is that improving SmartCare does not also necessarily mean improved surveillance analysis and use of SmartCare data for PMTCT purposes, which might need to be explored further. For example, authors might also look at how making ‘quality’ data available shapes use, and why. Moderate/minor:
--	--

	-text should be reviewed overall for language. E.g. pg 15 line 58 needs rewriting and there are several places like that. -‘Data is’ or ‘Data are’ – both used, need to be consistent (I would suggest plural usage) -proofreading needed throughout as well; e.g., pg 4 line 17 missing a period; the font used for the ‘Is’ on pg 13 in the quotation that begins “When I get time” (line 29) seems inconsistent; there are several of these problems -4.3.1 Data entry header should be in italics
--	---

VERSION 1 – AUTHOR RESPONSE

Reviewer Comment 1:

This paper presents results of a qualitative study on implementation of the SmartCare system for PMTCT programs in Zambia, based on 17 in-depth interviews with system stakeholders, 3 focus groups with program clients, and observations of system use.

Overall comments:

This paper is an important qualitative case study of the Zambia SmartCare system. There are relatively few in-depth qualitative studies of health information systems in low-resource settings, so this study is poised to make a contribution to the literature. The results presented are important and the overall conclusions seem very credible. The strongest gaps in the paper are: 1) the lack of a clear research question; 2) limited coherence in the justification of the sampling strategy; and 3) the lack of conceptual or theoretical grounding of the study analysis and results. Overall, stronger contextualizing of the study findings in the Discussion section and stronger writing could help push this manuscript to make a greater contribution to the literature. SmartCare seems to represent a “first generation” electronic health record in the Zambian setting, with some of the same weaknesses that have hampered other similar systems.

Greater linkage of the study findings to the literature on best practices in EMR design and implementation would be good.

Response1:

The authors acknowledge the compliments and the comments that will definitely improve the quality of the manuscript. The specific points are addressed in the responses to the detailed comments below.

Reviewer 1 Major comments

The authors state in the Introduction “The overall goal of our study is to improve the implementation of an EHR system in Zambia labelled ‘SmartCare’ and address the fundamental need for timely and high-quality data” (page 4, lines 50-54). This explains the motivation for the study, but not the central research question. This ambiguity is a problem throughout the manuscript. The study design and sampling strategy was also not clearly tied to a research question. Their purposeful sampling scheme lacked some coherence because they indicated that they sought sites with both stronger and weaker data quality. This suggests a research purpose to identify factors contributing to strong data quality in some sites and weak data quality in others. However, they also indicate that they included only Lusaka where data quality tended to be weaker, suggesting that they did not actually try to contrast factors in stronger vs. weaker sites, but rather sought to identify challenges in weaker sites only. The fact that they interviewed a range of stakeholders including clients suggests that perhaps their research purpose was to describe perceptions of the success of implementation of SmartCare from diverse stakeholder perspectives. In any case, a clearer statement of the research question would be helpful. The abstract includes a somewhat clearer statement of the study aim: “This study aimed to

investigate the challenges in implementing a Zambian EHR system labelled 'SmartCare' in order to improve PMTCT data."

Discussion page 17, lines 15-17: The Author's state: "The depth of the inquiry enabled us to consider a range of explanatory factors. Most of these factors were related to the EHR system and its implementation." However, their study lacks a clear research question -- they do not identify what they are trying to explain (explanatory factors for what?).

Response to major comment

The ambiguity of the study object has been addressed by indicating that it aimed to investigate the challenges in implementing a Zambian EHR system labelled 'SmartCare' from diverse stakeholder perspectives in order to improve PMTCT data collection so that it can be used for clinic performance strengthening and programme monitoring as our quantitative studies indicated that SmartCare data had quality shortfalls; see page 4 last paragraph.

Reviewer 1 Method comment

Methods: the authors should discuss how the 22 participants of the FGD were recruited, and any incentives offered. The authors should describe the setting from which the participants arose (if the FGD participants were recruited from 3 different clinics in Lusaka, for example).

Response: The FDG participants were recruited from three different facilities, two from Lusaka and one from the peri-urban area; this has been indicated in the manuscript on page 7, last paragraph.

Conceptual basis of the study. While it is certainly the authors' right to choose to come from a theory-free perspective in designing and carrying out the study ("Data were collected and analysed without a constraining overarching framework, which enabled us to capture the diversity of experiences of the SmartCare implementation efforts"), I found the lack of a conceptual basis to be a weakness in the presentation of the Results and the Discussion. In particular, I felt that in the Discussion section, the authors could have done a better job presenting and discussing their findings in reference to the broader literature. Examples of papers which might be of use in shaping the structure of the Discussion section include:

1. Fritz F, Tilahun B, Dugas M. Success criteria for electronic medical record implementations in low-resource settings: a systematic review. *Journal of the American Medical Informatics Association*. 2015; 22(2):479-488.
2. Khoja S, Durrani H, Scott RE, Sajwani A, Piryani U. Conceptual framework for development of comprehensive e-health evaluation tool. *Telemed J E Health*. Jan 2013; 19(1):48-53.

Following a conceptual framework or set of categories like that identified by Fritz or the KDS paper, might help in organization of the Discussion section.

Response to Method: The discussion section has been re-organized, as advised and the authors appreciate the examples of the papers suggested by the reviewer. We had also earlier referenced Fritz et al (2015) in our manuscript discussing the shortage of qualified information technology staff who are dedicated for EMR system.

Results: The client perspective is described only very thinly in the results (page 15, lines 25-33). this does not offer much insight on the patient perspective. I encourage the authors to see if there is any additional meaning or insight to be gained from the client FGDs.

Response: It is unfortunate that our FDG participants do not have insights on the implementation of the EHR. This is likely due the fact that the patients do not interact with SmartCare directly. It was also beyond the scope of our study to investigate how they relate with the clinicians as this could also

have had an impact on their perceptions. This point has also been included in the discussion on Page 17, second paragraph.

The recommendation on upgrading to a networked EMR (final paragraph of Discussion section) reflects a mismatch in understanding about the OpenMRS platform (see <https://openmrs.org/>) and about other EMRs deployed in low resource settings. Is the recommendation to convert the system to the OpenMRS data model and platform? Or to enhance SmartCare to encompass some of the features mentioned in the paragraph? It is not clear that the features mentioned would address some of the main implementation challenges the author has identified (too much data being collected, without value added for front-line personnel; mixed technical performance; lack of staffing; lack of feedback and data use).

Response: The paragraph has been re-written for clarity and highlighted on page 16, paragraph 2.

Limitations

A limitation of the study seems to be that it focused on implementation of SmartCare for PMTCT, but addressed issues which seem to relate to the entire system. Do the authors think that their findings would have been similar or different, had they also included stakeholders (health workers and clients) from other health services like TB or HIV ART services?

Response: The authors postulate that the results of the study would have been different if stakeholders from other health were included due the health facility setup where PMTCT services are located within the ANC department which usually far from the data entry rooms and general ART services. This point is addressed on page 17, last paragraph.

Reviewer 1 minor comments:

Introduction

The authors indicate they completed a systematic review of literature (page 4, line 25). Usually a systematic review is an end in and of itself, and involves citing the databases searched, the search terms used, and how the review was done. It's not clear to me that the authors followed a formal systematic review process. It might be best to drop the word "systematic".

Response: The term systematic has been dropped.

Sampling: before diving into the discussion of sampling on page 6, line 37, it would be best to reiterate the types of qualitative data collection and number of interviews done (i.e. move text from page 7, lines 16-19 to start the sampling section).

Please copy edit throughout. Please fix typos and grammar problems (e.g. page 7, line 30 refers to "stuff" rather than "staff").

Need to explain the role of IPs in Zambia's health system, when they are introduced in the manuscript. A reader without knowledge of PEPFAR-supported programs may not understand the phrase "IPs".

The hybrid data collection process described on page 11, line 50 – page 12, line 5 should be moved to the Methods section where the system itself is described.

Response: The sections on sampling and data collection have been reorganised to improve clarity and an introductory sentence added to the section (pages 6-7). The typos have been corrected, and the sentences moved as advised.

Figure 1: Authors should define abbreviations, like “HIA 1”, “OPD” etc.

Response: The abbreviations for figure 1 have been included in the figure legends on page 23.

Discussion page 16, line 47: “There was a notable lack of appreciation of the system”. The authors should clarify who had a lack of appreciation of the system. The authors could underscore the diverse purposes of the system as it was originally conceived and the lack of user-centered design processes during software development.

The authors present the following result without context, and fail to return to it in the Discussion section (page 13, lines 50-51): “The SmartCare Database system has reports that are built in the application. However all the partners alluded that these reports are very inconsistent and incorrect.” Is this simply a perception of end users, which is explained by poor data quality leading to reports that do not seem to reflect reality? Or is it an actual problem of faulty queries producing the automated reports, which relates to issues about lack of financial resources for sufficient design and testing of in-built reports in the system? Or do the authors have limited information to “unpack” this perception from end users. Giving some further indication of the nature of the problem, or at least discussing it further would be helpful.

Response: The observation reported by data entry clerks was that SmartCare reports did not match client numbers seen in the clinic (page 13). Data entry clerks provided some suggestions why that might be and data entry limitations discussed elsewhere could have contributed. However, we did not have access to the SmartCare reporting software so are unable to ‘unpack’ this observation further. We have added in our discussion a recommendation of establishment of SmartCare help desks on page 17, paragraph 2.

Abstract

The abstract results do not highlight the same issues as the results section in the paper. A major issue highlighted in the paper is the burden of information needing to be entered, which is not mentioned at all in the abstract. It might make sense to use the four (or three – see below) major findings issues listed in the paper to structure the abstract results section.

Response: We have added in the abstract that there is a burden of information that is not collected and entered into the database which has resulted in poor data quality.

Reviewer Comment 2:

Introduction

This paper begins by describing the benefits and barriers to using routinely collected EHR data for PMTCT surveillance purposes, with a focus on Zambia which uses the SmartCare EHR System. Barriers to use of SmartCare have been identified through a literature review and preliminary work, which focuses mainly on the quality of the EHR data, such as missing data components. The stated goal of the study is then to build on these reviews to “improve the implementation of the EHR system in Zambia and address the fundamental need for timely and high-quality data” (pg 4). To do this, the authors employ a qualitative approach of focus groups and interviews to presumably understand the issues related to use of the EHR data system.

There are primary two concerns here. The first is that the introduction does not state the investigative purpose of the study; one imagines that system improvement would be an outcome from some kind of empirical study, but of what? The abstract states this more clearly: “the study aimed to investigate the challenges in implementing...” This or similar explicit language about aims needs to be brought into the introduction to state the purpose of the study at present, not only its implications for future.

Response: The study purpose has been rewritten by indicating that it aimed to investigate the challenges in implementing a Zambian EHR system labelled 'SmartCare' from diverse stakeholder perspectives in order to improve PMTCT data collection so that it can be used for clinic performance strengthening and programme monitoring as our your quantitative studies indicated that SmartCare data had quality shortfalls; see page 4 last paragraph.

A second concern revolves around the degree to which the study can illuminate issues related to use of SmartCare for PMTCT monitoring purposes: likely a more important outcome of a study such as this, but one that is different from the focus on the SmartCare EHR system in general. The authors do not distinguish these different types or steps in implementation, and I would suggest attending to them each by clarifying what is meant in this paper by 'implementation' (as well as what questions cannot be answered by this study). They can then address this again by discussing the implications of results in the discussion and limitations sections – see some additional comments about this below.

Response: The implications of the use of SmartCare for PMTCT monitoring which differ from the focus on the SmartCare EHR have been discussed under the limitations of the study on Page 17-18. In this context the implementation of SmartCare was based on getting good data, later stages involving retraining of staff, reorganizing supply chains are not what we mean here by implementation.

Reviewer Comment 2:

Methodology

Sampling

"The approaches to recruitment of participants were flexible" – what does this mean? What was the negotiation? It is unclear how the recruitment actually occurred.

How were the patients recruited, all told 22? That's a big number; how were they selected, and how was representative ensured?

Response: The recruitment of in-depth interviews was flexible in the sense that we were not specific on the positions of the IP participants but it was based on either their experience in working with SmartCare, availability for interview, or some based on those who were chosen by the management of that particular IP. We also indicated in the manuscript that these were negotiated.

The recruitment of FDG participants was done by PMTCT coordinators, who were also instructed to balance the numbers of pregnant and lactating women. This information has been added on page 7, in the first paragraph.

Instead of figure 2, it would be helpful to include a summary table that breaks down the types of interviews by categories included on bottom pg 6, para beginning line 54. High/low performing; paper/computer. Later when citing quotations, it would be helpful to refer to the interviewee by clinic type.

Response: Figure 2 has been deleted and its information has been included in Table 1.

Analysis

The grounded description of analysis of 'no overarching frameworks' or 'pre-conceived ideas' does not seem plausible. Beyond that there is generally a strong applied steer in this kind of work; the authors state the goals of the study to explain specific "deficiencies" in optimizing the SmartCare system, so it is much more reasonable to assume that these areas guided initial as well as later coding. If a different 'unconstrained' approach was used, then cite to the method, justify and describe it more persuasively.

Response: The 'no overarching framework' has been removed and we have clarified that coding was also done on issues that were related to the reasons for missing PMTCT data. (Page 8, paragraph 4.)

Results

The four primary thematic areas/issues seem fine, though would suggest numbering them in the first results paragraph for clarity. When describing each issue, use of quotes needs better attention. A quotation needs to be set up, including part of a sentence rather than its own sentence (such as, "As X described, ".") and each quotation needs to be contextualized or explained. This may happen prior to a quote, but it often happens after. If a second quotation is used, it should present another idea or a nuance or elaboration of the first, and again the authors need to point this out for the reader rather than simply including a second excerpt. At present, sometimes two quotations side by side don't seem to be making the same point (e.g., top of page 10; how does PEPFAR performance relate to the range of clinical inputs?). At other times, the prose is unsupported by the text (bottom of pg 10). I advise authors to make each selected quotation more robust by explaining how to understand them, and giving more context to interpret them. Again, a summary table can then be used to show how each interviewee is located across salient categories.

The 'facility set up' issue 4 section is very thin, and perhaps can be incorporated into the 3rd issue, which seems very process oriented.

The quotations from the focus groups are also quite thin; if the patient interviews are not useful to this analysis, they should be removed. At present, they feel tokenistic, which is strange if 2 were completed with over 20 individuals so to sum up their contribution with 3 quotes that have nothing to do with data does nothing for the argument. It is understandable that patients may not have awareness of the more technical issues of data entry and use, so pulling the patient piece out may make sense.

Response: Some of the quotations have been separated, and highlighted. The authors also feel that even if the FDG participants do not have anything to do with data entry, their response must be represented since they are key stakeholders. We have also commented about them in our discussion.

Discussion

Consider citing to implementation science literature (e.g., diffusion of innovations – Greenhalgh et al 2004) about how organizations adopt new innovations, or new practices. It is gestured to in feasibility, but the authors could bring in other categories as well. When describing the importance of gaining buying among end users at bottom of pg 16, the discussion begs the question of how this is achieved, and turning to implementation science lit might help to offer some suggestions. Again, it would be great to talk about implications of findings not only for getting better data into the system, but using it for program monitoring – not the same activity!

Response: The Discussion has been reconstructed and several changes have been highlighted in the manuscript. We appreciate the implementation science literature suggested by the reviewer; we have included this useful reference.

Limitations

Though the paragraph states several limitations, only 1 is listed. What are the others?

One obvious issue (again) is that improving SmartCare does not also necessarily mean improved surveillance analysis and use of SmartCare data for PMTCT purposes, which might need to be explored further. For example, authors might also look at how making 'quality' data available shapes use, and why.

Response: Limitations of the study have been expanded as suggested.

Moderate/minor:

-text should be reviewed overall for language. E.g. pg 15 line 58 needs rewriting and there are several places like that.

-‘Data is’ or ‘Data are’ – both used, need to be consistent (I would suggest plural usage)

-proofreading needed throughout as well; e.g., pg 4 line 17 missing a period; the font used for the ‘Is’ on

pg 13 in the quotation that begins “When I get time” (line 29) seems inconsistent; there are several of these problems

-4.3.1 Data entry header should be in italics

Response: The typos have been corrected.

Thank you for your consideration of this revised manuscript.

VERSION 2 – REVIEW

REVIEWER	Nancy Puttkammer University of Washington
REVIEW RETURNED	26-Jun-2019

GENERAL COMMENTS	There remain some copy editing issues in the manuscript. While the authors state they have addressed all issues, I note several bothersome points: 1) authors sometimes use "data is" and sometimes "data are"2) authors sometimes use "electronic medical record (EMR)" and sometimes use "electronic health record"3) page 16, authors state: "In Ethiopia the quality of data were reported to be affected by dual documentation where both paper based and electronic systems were used (20)." The authors should make clear if quality of data was favorably or unfavorably affected? Please ensure consistency! In the sections added by the authors in yellow, the wording in a few places is awkward or lacks grammatical correctness. Requesting a fresh set of eyes for final copy-editing will help this piece.
---

VERSION 2 – AUTHOR RESPONSE

Reviewer(s)' Comments

Reviewer: 1

1) Authors sometimes use "data is" and sometimes "data are"

Response: All “data is” have been replaced with “data are”.

2) Authors sometimes use "electronic medical record (EMR)" and sometimes use "electronic health record"

Response: All “EMR” have been replaced with “EHR”.

3) Page 16, authors state: "In Ethiopia the quality of data were reported to be affected by dual documentation where both paper based and electronic systems were used (20)." The authors should make clear if quality of data was favorably or unfavorably affected?

Response: We have added a sentence to qualify the preceding statement "In Ethiopia the quality of data was reported to be affected by dual documentation where both paper based and electronic systems were used (20). In this study completeness of paper-based records was slightly better than electronic records."

4) Please ensure consistency! In the sections added by the authors in yellow, the wording in a few places is awkward or lacks grammatical correctness.

Response: The grammatical errors have been corrected.

Reviewer 2

Please ensure that all responses to the original comments from reviewer 2 have been incorporated into the manuscript, such as the clear definition of "implementation".

Response: We have added a statement on page 4 last paragraphs that explains that in this context the implementation of SmartCare was based on getting good data from individuals attending health facilities, the later stages involving retraining of staff, reorganizing supply chains are not included in this definition of implementation.

Thank you for your consideration of this revised manuscript.